# Estimating Carbon Sequestration Potential of Forest and Its Influencing Factors at Fine Spatial-Scales: A Case Study of Lushan City in Southern China

**DOI:** 10.3390/ijerph19159184

**Published:** 2022-07-27

**Authors:** Geng He, Zhiduo Zhang, Qing Zhu, Wei Wang, Wanting Peng, Yongli Cai

**Affiliations:** School of Design & China Institute for Urban Governance, Shanghai Jiao Tong University, Shanghai 200240, China; hegeng@sjtu.edu.cn (G.H.); zhangzhiduo12532255@sjtu.edu.cn (Z.Z.); jzzc0504@sjtu.edu.cn (Q.Z.); wwang1988@sjtu.edu.cn (W.W.); pw2020@sjtu.edu.cn (W.P.)

**Keywords:** fine-scale, biomass expansion factor, tree growth equation, carbon sequestration potential, site characteristics

## Abstract

Accurate prediction of forest carbon sequestration potential requires a comprehensive understanding of tree growth relationships. However, the studies for estimating carbon sequestration potential concerning tree growth relationships at fine spatial-scales have been limited. In this paper, we assessed the current carbon stock and predicted sequestration potential of Lushan City, where a region has rich vegetation types in southern China, by introducing parameters of diameter at breast height (DBH) and tree height in the method of coupling biomass expansion factor (BEF) and tree growth equation. The partial least squares regression (PLSR) was used to explore the role of combined condition factors (e.g., site, stand, climate) on carbon sequestration potential. The results showed that (1) in 2019, the total carbon stock of trees in Lushan City was 9.22 × 10^5^ t, and the overall spatial distribution exhibited a decreasing tendency from northwest to south-central, and the carbon density increased with elevation; (2) By 2070, the carbon density of forest in Lushan City will reach a relatively stable state, and the carbon stock will continue to rise to 2.15 × 10^6^ t, which is 2.33 times of the current level, indicating that Lushan forest will continue to serve as a carbon sink for the next fifty years; (3) Excluding the effect of tree growth, regional forest carbon sequestration potential was significantly influenced on site characteristics, which achieved the highest Variable Importance in Projection (VIP) value (2.19) for slope direction. Our study provided a better understanding of the relationships between forest growth and carbon sequestration potential at fine spatial-scales. The results regarding the condition factors and how their combination characteristics affect the potential for carbon sequestration could provide crucial insights for Chinese carbon policy and global carbon neutrality goals.

## 1. Introduction

As an integral component of terrestrial ecosystems, forest ecosystems are a massive global carbon reservoir [1]. Forests sequester 2/3 of the total terrestrial carbon sequestration annually [2]. They perform a critical and irreplaceable function in lowering the rate of accumulation of greenhouse gases in the atmosphere, which helps to mitigate global warming [3]. Since the 1980s, due to large-scale afforestation programs, forests in southern China have accounted for more than 65% of the national carbon sink [4,5], much higher than in northern regions. At the present stage, China’s strategic goal of “reaching a carbon peak by 2030 and achieving carbon neutrality by 2060” requires a focus on emission reduction and sink enhancement, so it is necessary to quantify the current carbon stock and sequestration potential, i.e., the maximum carbon capacity that can be stored in forest ecosystems without human interference [6]. For a thorough understanding of the role of forest ecosystems in the carbon cycle, an accurate calculation of the carbon sequestration potential of forest ecosystems is required. Not only does it aid in quantifying the impacts of forests on global warming, it also aids forest management decision-making processes [7,8].

Existing methods for estimating carbon sequestration potential are shaped by the extension of carbon stock estimates. The estimation of carbon stocks using the biomass expansion factor (BEF) is considered relatively reliable [9], which determines the forest biomass and forest volume as a fixed ratio and estimates the carbon stock of the region by the mean ratio method (MRM) [10]. The continuous BEF method was proposed by Fang et al. and was used to estimate the carbon stock of forests in China [11]. The forest carbon sequestration potential is the difference between the maximum forest carbon capacity and the current (or a given year) forest carbon stock. Since the carbon density of mature forests can represent the maximum carbon density of forests in similar regions, the carbon stock at this time is frequently assumed to be the maximum forest carbon capacity [12]. In the natural state, the carbon stock of forest vegetation usually increases rapidly with the increase of forest age (successional stage), then slows down and reaches a steady state [13]. This increasing trend, described as S-shaped, was also reported by Taylor et al. [14] and Rothstein et al. [15]. The carbon stock of existing forests in China increases with the age of the forest, and all types of forests at different age stages can sustain carbon sequestration [12].

Current studies have introduced age into the estimation, using the relationship between biomass density and tree age to estimate carbon sequestration potential. Mostly used for large-scale study areas, such as the whole of China [16] and Finland [17], the estimation method has been thoroughly developed. Due to the large geographical span, diverse climate types, and complex tree growth in the large-scale areas, it is feasible to use this simplified connection to estimate carbon sequestration potential as a reference value. However, for the fine-scale regions, such as the county-level study areas in Hebei Province [18] and Tibet Autonomous Region [19], the carbon density and carbon sequestration potential of forest vegetation in 2050 were estimated by directly fitting the biomass-forest age relationship using the biomass converted from storage volume, ignoring the fact that the change of forest stock volume is disturbed by various conditions and is an artificial estimate during the survey [20]. This does not accurately reflect the growth of trees, and the estimation for this scale is still questionable, affecting the actual forestry carbon sink project design. As a result, we take the more accurate depiction of tree growth as an entry point. According to the widely established model for estimating storage volume, i.e., the binary standing volume model, DBH and tree height can visually represent the growth of volume [21], which can be combined with the tree growth equation [22] to reduce the uncertainty in estimation. The stochastic simulation is used to more accurately represent the change in accumulation volume during the growth of trees and to improve the accuracy of estimating forest carbon sequestration potential.

The forest carbon sequestration potential is not only influenced by forest growth but also by climatic factors [23], topographic factors [24], land use change [25], management measures [26], etc. Since the carbon stock of forest ecosystems is the fundamental parameter for studying the carbon exchange between forest ecosystems and the atmosphere [27], these existing studies mostly choose the current state of carbon stock as the variable and analyze its influencing factors, ignoring the growth status of the forest and focusing solely on the impact of environmental conditions under the current state of carbon stock [24,28]. The current carbon stock is influenced by age group composition and dominant species type, so the carbon stock is often not stable [12]. The carbon sequestration potential is the maximum possible growth of forest carbon stock under the current scenario, which is the predicted result after the dynamic growth of the forest. At this time, the average age of all tree species has reached the mature forest stage, and the carbon stock is relatively stable, which is convenient to reveal the relationship between the carbon sequestration potential and the current condition factors in the study area. Furthermore, most previous research has concentrated on single components such as elevation factor, canopy density, rainfall factor, and so on [23,24,28], and less attention has been paid to the combination characteristics among factors. The forest growth condition of the carbon sequestration potential is used in our study as an entry point to analyze the effect of single factors of the condition factor on forest carbon sequestration. Based on this, we also try to combine single factors of the same type and examine the magnitude to which influence on carbon sequestration potential among various combined features, this will assist in eliminating the interference of uninterpretable information such as multiple correlations and better analyze the influence of multiple condition factors such as site, stand, and climate in a comprehensive manner.

Given the above, Lushan City in southern China, a “natural laboratory” for studying forest ecology [29], was selected as a case region of the study. The specific objectives of the study were to (1) better understand the relationships between forest growth and carbon sequestration potential at the fine-scale study area; (2) estimate the region’s carbon sequestration potential by analyzing the current characteristics of carbon stock; and (3) reveal the influence of condition factors and their combination characteristics (site, stand, and climate) on carbon sequestration potential without forest growth disturbances.

## 2. Materials and Methods

### 2.1. Study Area

Lushan City (115°49′42″–116°8′18″ E, 29°9′6″–29°38′32″ N) is located in the north of Jiangxi Province, with a total area of 764.54 km^2^ (Figure 1). The city, in the East Asian monsoon region, has a humid subtropical climate influenced by both Lushan Mountain and Poyang Lake. The annual average temperature is 15.3~17.3 °C, the precipitation is uneven in all seasons, and the dry and wet seasons are obvious. Lushan City owns Lushan Nature Reserve, which has been operating for 41 years since 1981 and has a subtropical forest ecosystem as its main conservation object. The forest coverage rate has increased from 42.00% before the establishment of the nature reserve to 80.70% at present, and the forest resources are abundant and well-maintained, with a wide diversity of tree species [29,30]. The total area of arboreal woodland in Lushan is 274.47 km², consisting of (i) natural forests (formed by natural underplanting, artificially promoted renewal or sprouting after a disturbance such as natural forest harvesting) and (ii) planted forests (formed entirely by machine seeding or artificial sowing, such as seedling planting, seeding and fly sowing). Among them, the area of natural forest is 201.30 km^2^ (73.34%), and the area of planted forest is 73.17 km^2^ (26.66%). Based on the main dominant species of the forest fine patches in the forest management inventory, the forest patches were classified into 7 types: *Pinus massoniana*, *Pinus taiwanensis*, and *Pinus elliottii* constitute pine forest (PF); *Cunninghamia lanceolata* and *Cryptomeria japonica* constitute Chinese fir forest (CFF); *Cinnamomum camphora*, *Quercus* L. and other hard broad species constitute broadleaf hardwood (BLH); *Populus* L., *Paulownia fortunei* and other soft broad species constitute broadleaf softwood (BLS); and mixed coniferous forest (MCF), mixed broadleaf forests (MBF) and mixed conifer-broadleaf forests (MCBF). Among them, PF (41.86%), MCF (18.30%), and CFF (17.95%) accounted for a higher percentage. According to the age of trees, the patches of forests in Lushan are mainly young and middle-aged, with the majority of trees between 20–40 years old (59.33%) and very few patches with an average age of more than 60 years old (0.72%). The complex and varied mountainous landscape of Lushan presents an elevation difference of about 1465 m. The vegetation shows more obvious vertical distribution characteristics, and 81% of the forest patches have an elevation greater than 100 m. Furthermore, initiatives including closing hills for afforestation, rehabilitating degraded forests, and tending to forests have been taken seriously in Lushan City to increase their capacity as carbon sinks (For example, the above projects involved 400 ha of forest in 2019). In conclusion, Lushan City has a diverse range of forest types and a considerable mountain microclimate, with the typical characteristics of subtropical mountain forests in southern China.

### 2.2. Data Sources

Forest management inventory is an important basic task for understanding the current state of forest resources and the ecological environment, providing a foundation for a scientific formulation of forestry development planning. The base data for this study was obtained from the forest management inventory data (FMID) in Lushan City, Jiangxi Province. Excluding economic forest, shrub forest, bamboo forest, and other forest patch types (The above types are incomplete in FMID), there were 5162 forest patches. After removing invalid data, 5077 valid forest survey patches were obtained. The survey recorded 76 forest class factors, including (i) site conditions (e.g., average elevation, slope direction, slope gradient, soil thickness, etc.); (ii) stand characteristics (average tree age, average DBH, average tree height, canopy density, etc.); and (iii) evaluation factors (forest naturalness, stand protection class, etc.), which provide sufficient variables and considerations for modeling. The survey accuracy of the sampled volume and forest patch factors is ensured through systematic sampling, and the overall regional volume accuracy reaches 80–85%; the error of the average tree height does not exceed 10%, and the error of the average DBH does not exceed 1 cm; the allowable error of the average age of natural forests is less than one age class period (10 years for PF, BLH, 5 years for CFF, BLS, mixed forests depending on the actual species composition), and the average age of planted forests is basically error-free. Furthermore, the FMID was completed in 2019, and we traveled to the field in September 2021 to conduct research and select some typical sample sites to verify the data’s legitimacy.

Climate data were obtained from nine meteorological stations in and around Lushan city from 2014 to 2018 daily rainfall and average temperature data from the China Meteorological Science Data Sharing Service (http://data.cma.cn/, accessed on 15 June 2022). Radiation data were high-temporal (3 h) surface solar radiation data from 2014 to 2017 sunshine hours in the Lushan area from the National Qinghai-Tibet Plateau Scientific Data Center [31] (http://www.tpdc.ac.cn, accessed on 15 June 2022). Data points were extracted using the fishnet extraction tool and then spatially interpolated using the inverse distance weighting (IDW) approach in ArcGIS Pro2.5 [32] to create the grid data of multi-year average temperature, precipitation, and radiation. We compared the effect of IDW interpolation with other spatial interpolation of climate data in the Poyang Lake basin study [33] and finally chose to use the result of the IDW for influence factor analysis. All raster spatial resolutions were unified at 30 m, and the projection coordinate system was unified at CGCS2000_3_Degree_GK_Zone_39.

### 2.3. Methods for Estimating Carbon Sequestration Potential

#### 2.3.1. DBH-Tree Height Growth Model

There is an obvious positive correlation between tree standing volume and its DBH and tree height [11]. We used the binary standing volume model, which has sufficient accuracy and is the most widely used [21], to describe the functional relationship, as shown in Equation (1).
(1)$$V=a_{0}D^{a_{1}}H^{a_{2}},$$
where V is the stumpage volume (m^3^), D is the average DBH (cm), H is the tree height (m); $a_{0}$, $a_{1}$, $a_{2}$ are the parameters to be fitted. The FMID were counted by forest patches, and 3–5 standard trees of the dominant species were selected in each forest patch for measurement, and the average DBH (D) of the cross-sectional area was used as the DBH data, and the average tree height (H) was used as the tree height data. The classifications were fitted to $a_{0}$, $a_{1}$, and $a_{2}$ to obtain model parameters that better fit this study area.

Changes in tree DBH and tree height are distinctive features of the performance with increasing tree age, and we selected samples of forest patches with similar natural conditions, divided into age groups, and proposed a simplified model of DBH and tree height growth. After a sufficient number of data samples passed the Shapiro-Wilk normality test [34], it can be assumed that the mean DBH and mean tree height of forest patches of the same mean age follow a normal distribution, and the trend of the model normal distribution parameters with mean age can be studied further. The relationship between DBH-tree height and age is difficult to construct with a uniform expression, so an attempt was made to employ the existing growth models Gompertz, Logistic, Korf, Mitscherlich, and Richards growth functions [22,35]. The above models were used to fit nonlinear curves for the expectation of DBH-tree height and their age, respectively. Using the highest R² and lowest RMSE as the test criteria, the best growth model for the forest type in this region was determined to be the logistic [36], which was selected for subsequent analysis, as shown in Equation (2).
(2)$$Y=\frac{c_{0}}{1+c_{1}e^{-c_{2}\cdot T}},$$
where Y is the DBH or tree height, T is the age of the tree, e is the natural exponential, $c_{0}$, $c_{1}$, and $c_{2}$ are the parameters to be fitted. This equation describes a three-parameter S-shaped growth curve, with $c_{0}$ showing the exact upper boundary of growth and $c_{1}$, $c_{2}$ jointly determining the growth rate of the curve, which is an ideal population growth model with important ecological significance and is widely used.

#### 2.3.2. Stochastic Simulation of Volume Growth

Under the premise that the mean DBH and tree height of the same mean age forest patch obey normal distribution, respectively, the mean volume of the forest patches should satisfy some joint probability distribution function of DBH and tree height according to the binary standing volume model (Equation (1)). Since the form of the distribution obtained from the solution of this function is complicated, it is not conducive to practical application. Therefore, we use MATLAB R2020b (9.9) and Origin 2019b to conduct a stochastic simulation. Based on the age series and the DBH-tree height growth function, the samples of DBH and tree height were drawn reflecting normal distribution. Further, we got a sample matrix of volume. The sample means were used as point estimates, leading to the expectation of volume under different forest types and ages. By stochastic simulation of volume growth, we gained a more accurate fit to the logistic growth function. More details about stochastic simulation can be found in Appendix A.

#### 2.3.3. Estimation of Carbon Sequestration Potential

Tree biomass density was significantly and linearly positively correlated with volume density [11] (Equation (3)), and forest carbon stock estimates were derived by multiplying forest biomass by the amount of elemental carbon in the biomass (i.e., the carbon content factor). Carbon density is the amount of carbon stored per unit area of forest biomass.
(3)$$W=\beta_{1}\cdot V+\beta_{0},$$
where W is the biomass density (kg/ha), V is the volume density (m³/ha), $\beta_{1}$, $\beta_{0}$ are model parameters, mainly based on the forest type conversion model proposed by Fang et al. [11] and Zeng et al. [37]. Due to the different tree species composition, age, and population structure of different vegetation types [38], the carbon content conversion coefficients may vary greatly. In this study, forest carbon stocks were measured based on the carbon content coefficients of each tree species (group) in the “Guidelines for carbon sink measurement and monitoring in afforestation projects” issued by the State Forestry Administration and previous research results [39,40] (Table 1).

Carbon sink capacity indicates the ability of vegetation to fix carbon per unit time, expressed as the increment of carbon stock in a certain time (Equation (4)). When the carbon density of the forest is relatively stable, the carbon sequestration potential is the difference between the carbon stock tending to the maximum and the carbon stock in the current year.
(4)$$CS=C_{t}-C_{t-1}=\gamma \cdot (W_{t}-W_{t-1}),$$
where the annual carbon sink CS is the difference between the corresponding carbon stocks of $C_{t}$ and $C_{t-1}$ in adjacent years and is equivalent to the product of the carbon content factor and the biomass. Equations (3) and (4) were implemented on Origin 2019b software.

### 2.4. Influencing Factors Analysis Method

In this study, the PLSR was used to explore the conditional factors of carbon sequestration potential [41,42] to effectively remove the interference of non-interpretative information. The results of carbon sequestration potential values were used as dependent variables to analyze the influence of single trait factors. A combination of single traits of the same type was attempted to construct the combined traits separately (Table 2) to reflect the degree of influence of a certain factor type comprehensively.

Multiple correlation diagnostics were first performed to calculate the variance inflation factor (VIF). It is generally considered that when VIF > 10, multiple correlations among the factors will seriously affect the estimates of partial least squares. After testing, all single and combined factors satisfy VIF < 10, indicating no significant linear correlation between the factors and can be used for PLSR. The factors that passed the diagnostic were selected for PLSR, and Variable Importance in Projection (VIP) was calculated to indicate the degree of explanation of the standard deviation by the factors, as shown in Equation (5).
(5)$$VIP_{j}^{2}=\frac{q\sum_{i=1}^{m}r^{2}(Y,x_{i})w_{ij}^{2}}{\sum_{i=1}^{m}r^{2}(Y,x_{i})},$$
where *q* denotes the number of variables involved in the analysis, m denotes the number of iterations, and in the ith iteration, $r(Y,x_{i})$ is calculated as the correlation coefficient between the dependent variable and all variables. $w_{ij}$ is the weight of variable j, which reflects the degree of explanation of the variables in the model. The sum of squares of VIP values of all variables is equal to 1. Factors with VIP < 1 are considered to have a low degree of explanation of the model, and factors with VIP ≥ 1 have a high degree of explanation. All the above tests were performed with Python 3.9.

## 3. Results

### 3.1. Modeling Results of Tree Forest Volume Growth

The fitted parameters of the binary standing volume model and growth simulation for each type of forest in Lushan City are shown in Table 3. The fitted parameters of the binary standing volume model and the DBH-Tree Height growth model generally had R² values above 0.90, which were well fitted, as shown in Figure 2. For each age group of different forest types, the growth function of volume expectation with tree age was fitted. It was found that the fitted logistic curves using continuous derivable logistic curves yielded good fitting results for the volume expectancy as a function of mean age. For the curves of relative tree height, relative DBH, and forest volume with age for specific forest types, please refer to Figure A1 in Appendix B. The fitted models for the major forest types showed statistical significance at the 0.01 level, and the R² values close to 1 confirmed the good applicability of the model for estimating tree forest volume in Lushan. This volume growth model illustrated the relationship between the volume of a dominant species forest with age in a simplified form, which provides a good basis for the estimation and prediction of carbon sequestration potential.

### 3.2. Characteristics of the Current Carbon Sequestration Capacity of Tree Forests

The average carbon density of tree forests in Lushan City in 2019 was 33.59 t/ha. The current state of carbon density showed an overall distribution pattern of high in the northwest and low in the south, decreasing from north to south, as shown in Figure 3. The carbon density contribution of different age groups of forest types at various altitudes was analyzed. Forest patches were more distributed at 0–100 m and 100–300 m altitudes, and the carbon density was 26.41–28.97 t/ha here, which was lower than the average carbon density of the study area (33.59 t/ha). The carbon density increased with elevation in the four gradient intervals higher than 300 m, closely related to the forest types at various elevations. The main contributing forest types for carbon density were CFF and MCF in the 300–600 m and 600–900 m gradient intervals. PF was the most significant contributory species in the other four gradient intervals, particularly in the highest elevation interval (1200–1465 m), where PF accounted for a considerable proportion under all conditions.

In 2019, the volume of forest storage in Lushan City was 2.34 × 10^6^ m³, the biomass was about 1.73 × 10^6^ t, and the total carbon stock was 9.22 × 10^5^ t (Table 4). The carbon stock indicates the overall state of a forest type, and the percentage of carbon stock contributed by each type of forest varies. Among them, the four forest types of PF, CFF, MCF, and MCBF provided 86.12% of carbon stock, with PF mainly providing 33.39% of carbon stock. The annual carbon sink of the Lushan forest was 3.02 × 10^4^ t from 2019 to 2020, and its main contributing sources were PF (40.66%), CFF (15.39%), and MCF (19.50%). The average carbon density of tree forests in Lushan had grown about 1.10 t/ha/a with a growth rate of 3.28% from 2019 to 2020. The lowest growth rate of BLS was 0.74 t/ha/a with a growth rate of 1.84%, and the highest growth rate of MBF was 1.35 t/ha/a with a growth rate of 3.41%.

### 3.3. Predicted Carbon Sequestration Potential of Tree Forests

The relationship between the carbon density of forests and tree age was examined based on the distribution of current forest age groups. A significant increase in carbon density will experience in the next 20 to 50 years, and it will achieve a stable state after 50 years. This relationship indicates that the upper limit of carbon density will be between 55 and 75 years (The year here refers to the average age of the forest stands). The carbon density of Lushan City will reach a relatively stable state in 2070, achieving the maximum carbon sequestration potential in the study area (Figure 4). The change of overall carbon stock in tree forests from 2019 to 2070 shows an upward trend of decreasing growth rate: The carbon stock continuously will increase from 9.22 × 10^5^ t to 2.15 × 10^6^ t, and the overall carbon density will raise from 33.59 t/ha in 2019 to 78.33 t/ha in 2070, increasing to 2.33 times of the original one. The potential carbon sequestration is about 1.23 × 10^6^ t, with a higher contribution from PF and MCF (Figure 5). PF has the highest carbon sequestration potential because of its absolute dominance of the land area. The annual carbon sink of tree forests shows a trend of increasing and then decreasing: the highest annual carbon sink will occur in 2030 with 3.39 × 10^4^ t, and will decrease to 8.06 × 10^3^ t in 2070 (Figure 5). The peak yearly carbon sink of diverse dominant species forest patches occurs in different years due to the varied tree species structure and age composition. Among them, the peak annual carbon sink of CFF is the earliest, reaching the maximum in 2019; the peak annual carbon sink of MCF is the latest, reaching the maximum in 2039; the remaining dominant tree types will reach the peak annual carbon sink in 2022–2032.

The carbon sequestration potential of natural forests is significantly higher than that of planted forests (Figure 6). And the carbon stock of natural forests is about 2.31 times higher than that of planted forests in 2019, increasing to 3.15 by 2070. The annual carbon sinks in planted forests will peak between 2025 and 2026, while that of natural forests will peak between 2031 and 2032. The growth rate of carbon density in natural forests is also consistently higher than that in planted forests, with both reaching the same level between 2035 and 2036. By 2070, the carbon density of natural forests will reach 80.03 t/ha, higher than that of planted forests at 69.99 t/ha, indicating that natural forests can provide a more effective carbon sequestration function for Lushan City.

### 3.4. Exploration of Factors Influencing Carbon Sequestration Potential

As shown in Figure 7, we analyzed the single factors of all samples, in which the VIP values of slope direction (2.19), slope gradient (1.24), and soil thickness (1.02) were greater than 1. Slope direction (SD) had the highest importance, indicating that the carbon sequestration potential was significantly influenced by site characteristics. Adding the combination factors for analysis, the VIP value of stand characteristics was 1.29 based on the original key factors. All were higher than their three single factors (forest density (1.28), vegetation cover (0.44), and canopy density (0.42)), indicating that the combination of stand characteristics had stronger explanations than the single factors. The effect of combined factors of site conditions and climatic factors was average and less important than some single factors. When the effect sizes of the combined factors were compared, the explanatory effects of both stand characteristics (1.27) and climatic factors (1.16) were larger than 1, with the explanatory effects of stand characteristics being stronger than those of site characteristics.

Furthermore, the parameters impacting carbon sequestration capability varied depending on the forest type. The VIP values of single factors of natural forests and the overall regional forests were not significantly different, and their key factors were all slope direction factors in site characteristics. It is worth noting that the influencing factors of carbon sequestration potential of planted forests are different from the overall regional forests, and the VIP values of two factors, soil thickness (1.67) and vegetation cover (1.42), are greater than 1. Regarding the combination of characteristics, the climatic characteristics of both natural and planted forests had stronger explanatory effects than the single factors. Comparing the effect sizes among the combination characteristics, climate characteristics had higher explanatory effects on natural and planted forests, respectively. The explanatory role of site characteristics on the carbon sequestration potential of natural forests was high (1.18).

## 4. Discussion

### 4.1. Estimation Methodology and Estimation Results

The biomass-forest age relationship has become a frequently utilized method for predicting future forest carbon pools and estimating forest biomass carbon stocks [16,43,44]. Existing research has mostly employed biomass-forest age connections to estimate carbon sequestration potential in larger-scale study areas, such as national [16] and provincial [45] scales, as well as incorporating stand age in the present stand growth model framework to reduce estimation bias [46]. However, unlike the predictive growth equation with DBH and tree height factors, Liu et al. [20] and Zhou et al. [47] showed that, while many studies have reported successful applications of fitting biomass-forest age relationships directly using biomass converted from forest volume [48,49], there are still questions about the accuracy and precision of volume estimates, particularly concerning reducing the uncertainty of model parameters [50]. As a result, explicitly fitting the biomass-age relationship using biomass transformed from volume fails to appropriately depict tree growth [51]. Since the actual forestry carbon sink projects are frequently carried out for fine scales, more accurate forecast results of carbon sequestration potential are required. In this study, considering the complexity of the forest survey samples in the fine-scale study area, the tree growth equation was re-fitted based on the characteristics of the binary standing volume model in which the DBH and tree height can visually represent the growth of storage volume, using the relationship between DBH, tree height, standing volume storage, and tree age. Compared with the original model, the model constructed in this study further refines the relationship between volume and age of a forest type using a stochastic simulation process, which can be applied even with limited forest biomass data and forest age observation, and provides a reference for the prediction of forest carbon sequestration potential at fine-scale regions.

In addition, our estimating results are consistent with previous research literature [52,53,54]. Excluding differences in the age structure of the study area and study methods, they are generally consistent with the results of previous research on carbon density estimation in Jiangxi Province (Table 5). Compared with carbon density estimates at the same study scale in Jiangxi Province, the average carbon density in Lushan City estimated in this study was higher than the carbon density in Taihe County estimated by Wu et al. [55], and the carbon density in Xingguo County estimated by Li et al. [56]. A possible reason is that our study was investigated 16 years later than those two, during which the tree forest maintained stable growth and carbon density continued to increase. The average carbon density in Lushan City is close to 36.0 t/ha, which was estimated by Zhang et al. [24] in the whole of Jiangxi Province. Compared with the carbon density in Jiangxi Province estimated by Li et al. [52] and Wu et al. [53] (23.87–27.2 t/ha), the average carbon density in Lushan City is slightly higher, and its contribution to the forest carbon sequestration function in Jiangxi Province is greater. Compared with the predicted carbon sequestration potential of arboreal forests based on biomass-age relationships in previous literature, the results were similar to those predicted by Wu et al. [55] and Qiu et al. [54], indicating the reasonableness of the model. Additionally, compared with the national data, the carbon density in Lushan City in 2020 is lower than the 50.51 t/ha predicted by Zhang et al. [46] and the 59.8 t/ha predicted by Xu et al. [16], which may be mainly because the tree forests in Lushan City are dominated by middle-aged and young forests, and the forest management in Jiangxi Province is primarily rough management with slow growth [57]. The predicted carbon density in Lushan City in 2050 is close to the predicted values for the national forest carbon density in 2050, which indicates that the forest vegetation in Lushan City has significant potential for carbon sequestration.

### 4.2. Factors Influencing Carbon Sequestration Potential

Based on the growth of forest age of different forest types, we quantified the future carbon sequestration potential of Lushan City forests. After incorporating information on stand developmental stages into predicting future forest carbon sequestration potential, this study found that forest carbon stocks accumulated rapidly at young ages and gradually saturated at later stages, which is consistent with He et al. [43,59]. After changes in forest carbon density have stabilized, mature and over-mature forests can also continue to accumulate carbon as stand age increases [60], and still hold a crucial role in the carbon cycle despite decreasing growth efficiency. Therefore, the carbon sequestration benefits given by forests as they grow and expand are ongoing. In addition to forest growth and development, forest carbon sequestration capability is intimately tied to large-scale afforestation and regional extension of ecological restoration efforts. In the next five decades, ecological restoration programs and sustainable forest management in China will increase forest area and biomass carbon intensity, making forests of various ages a carbon sink [46]. And according to the China Forestry Sustainable Development Strategy Research Group, the quantity and quality of China’s forests are expected to enter a phase of steady development, which implies that the capacity of increasing forest carbon sequestration potential may be limited. As a result, to acquire better forest carbon sequestration potential assuming normal forest growth and development, it is required to investigate the influence of condition factors on carbon sequestration potential.

The predicted carbon sequestration potential value was used as the dependent variable in this study. The site characteristics had a significant impact on carbon sequestration potential, with slope direction having the most impact, which was significantly and positively correlated with the value of carbon sequestration potential. This result is consistent with the previous regional research findings in Jiangxi Province. Wu et al. [61] examined the vegetation carbon density of major forests in the Poyang Lake basin. They discovered that slope direction and gradient had a substantial impact on vegetation carbon density. Since the slope direction, slope gradient, elevation, and other site features have redistribution effects on surface light, heat, and water resources, which affect the forest growth and, consequently, the carbon pool. The findings imply that the research area’s carbon sequestration capacity is greatly influenced by the azimuth of solar irradiation, and the sunny slope (i.e., south slope) may yield stronger carbon sequestration [24]. 

The key factors influencing the carbon sequestration capability of various origins’ forests are diverse, resulting in various management strategies. Natural forests and the overall forests in the region have comparable crucial features, and they are all tied to site characteristics. The protection of natural forests should be encouraged, and the slope direction and slope gradient should be emphasized in the implementation of natural forest protection projects, which will avoid the reduction of forest carbon sink capacity caused by problems such as soil erosion. On the other hand, the key factors of planted forests are soil thickness and vegetation cover. Relatively thicker soil and relatively higher vegetation cover can provide a higher carbon sink. Therefore, when predicting the carbon sequestration potential of planted forests in the future, the above factors can be considered as the main control factors for modeling to improve the prediction accuracy. To provide favorable conditions for the expansion of carbon sink in a planted forest, more consideration should also be given to the aforementioned components when developing planted forest initiatives. Furthermore, when the findings of the multifactor combination were compared, the climatic combination had a greater impact than the site and stand characteristics. The growing season was effectively extended by the rises in temperature and precipitation, which also increased microbial activity, photosynthetic capacity, and plant growth and respiration [62]. This improved the capacity of forests to store carbon [5]. Therefore, the climatic combination characteristics can be considered to incorporate into the prediction model, allowing multiple climate condition scenarios to be established to more correctly estimate the future carbon sequestration potential of forests.

### 4.3. Uncertainties and Potential Constraints

Carbon stocks in forest ecosystems are primarily influenced by two aspects. On the one hand, changes in forest biomass and the accompanying changes in the carbon cycle, and on the other hand, changes in the forest soil carbon pool, namely the balance between imports and losses of organic carbon into the soil [9]. Solar radiation also plays an important role in plant carbon sequestration. For example, sunny slopes can lead to strong soil mineralization and evapotranspiration, which may limit plant carbon sequestration. The estimation of carbon sequestration potential is somewhat biased because actual measurements of soil nutrient mineralization and evapotranspiration have not been carried out. Due to the lack of data on understory vegetation, herbaceous layer, deadwood layer, dead wood, and soil layer in the FMID, this study did not cover the carbon stocks of the categories mentioned above and only considered the carbon stocks of live trees, so the estimation of forest ecosystem carbon stocks in Lushan City was quite underestimated.

The predictions in this study are also based on certain assumptions, which lead to some uncertainties in the results: first, the maximum carbon sequestration potential is an estimate based on spatial and temporal intergeneration, assuming no forest disease or mortality, and that existing forests grow naturally according to the growth equation, which only represents the maximum potential that a forest type or age can achieve under ideal conditions. In actuality, forests are affected by disease and mortality during the growth process, which may result in exaggerated estimations of carbon sequestration potential [38]. Second, if China’s forestry development and forest cover expand, the fraction of newly generated forests may fluctuate in the forecast process [6]. On the other hand, there are high uncertainties in the tree species composition and age groups of newly created forests, which may lead to inaccurate prediction results [58]. Hence, the newly created forests are not included in the estimation, and the prediction of carbon sequestration potential is slightly underestimated.

Finally, the impacts of anthropogenic and natural disturbances on forest carbon sequestration were not considered. With the increasing emphasis on forest protection through regulations such as “peak carbon dioxide emissions and carbon neutrality”, it is reasonable to expect that human activities such as logging will cause minimal direct disruption of natural forests in the future [6]. However, for the disturbance of planted forests under the influence of various anthropogenic activities (e.g., afforestation, logging, irrigation), the future carbon sequestration potential of forests still varies greatly [16,23]. Factors such as climate change, elevated atmospheric CO_2_ concentration, and nitrogen deposition may also affect the accumulation process of forest biomass density, and estimating forest carbon sequestration capability based on current climate circumstances may also introduce some uncertainty [23,54]. A more comprehensive study, including climate changes such as warming and drought, as well as the effects of other anthropogenic disturbances on future forest carbon sequestration, should be conducted.

## 5. Conclusions

Our study provided a better understanding of the relationships between forest growth and carbon sequestration potential at fine spatial-scales by introducing BEF and tree growth equations. Moreover, we further explored the effect of the combination of factor characteristics on the carbon sequestration potential, excluding forest growth effects, which provides crucial insights for Chinese carbon policy and global carbon neutrality goals.

By 2070, the carbon density of forests in Lushan City will reach a relatively stable state, and its carbon stock will be close to the maximum, indicating that Lushan forests will serve as a long-term carbon sink in the next fifty years. Among them, pine forests and mixed coniferous forests have a higher carbon sequestration contribution. In addition, the carbon sequestration potential of natural forests was much higher than that of planted forests, with the gap widening as the woods aged. Thus, conserving natural forests should be encouraged to sustain carbon sequestration capacity in future afforestation projects, and replantation site characteristics should be carefully considered in the afforestation projects to increase carbon sequestration capacity. Slope direction, slope gradient, soil thickness, and vegetation cover factors are important factors of forestry carbon sink, which should be paid attention to in implementing forestry carbon sink projects.

More importantly, incorporating DBH and tree height data from the binary standing volume model can better represent forest growth changes. A stochastic simulation process could be used to further refine the relationship between the standing volume of forest types and the age of the trees, which improved the accuracy of the prediction of carbon sequestration potential at the fine-scale areas. It can also be applied in the case of limited forest biomass data and stand age observation, enriching the ways of predicting forest carbon sequestration potential. Future work should also consider climate changes on future forest carbon sequestration for better achieving global carbon neutrality goals.

## Figures and Tables

**Figure 1 ijerph-19-09184-f001:**
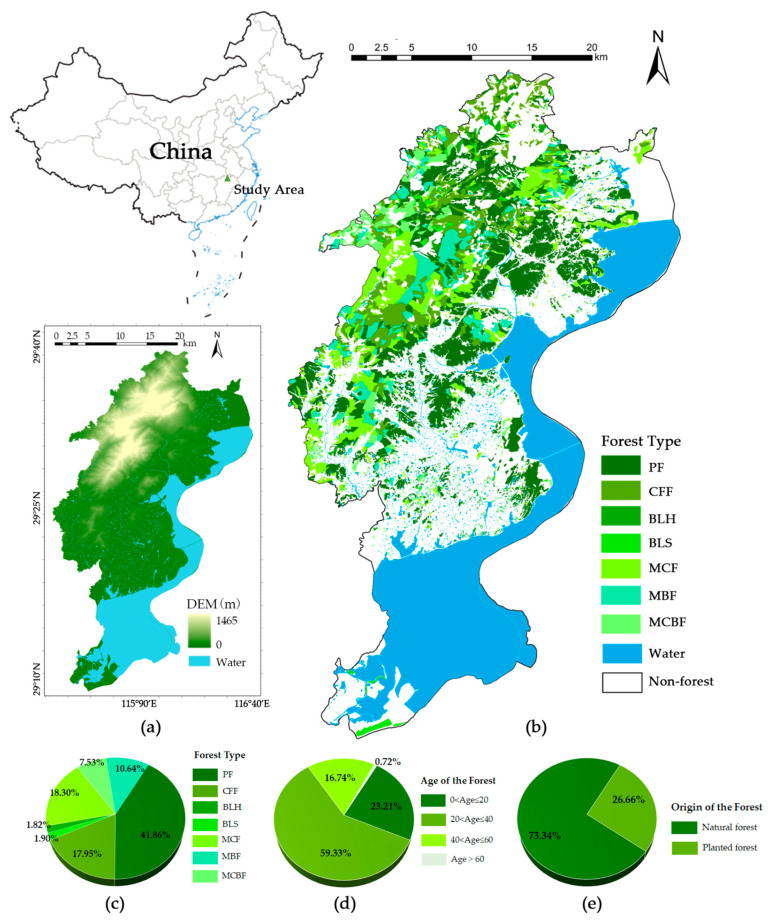
Location, topography, basic forest information, and forest patch distribution in the study area. (**a**) shows the location and topography of the area; (**b**) shows the distribution of different forest types. Based on the main dominant species of the forest fine patches in the forest management inventory, the forest patches were classified into 7 types: PF refers to pine forest composed of *Pinus massoniana*, *Pinus taiwanensis*, and *Pinus elliottii*; CFF refers to Chinese fir forest composed of *Cunninghamia lanceolata* and *Cryptomeria japonica*; BLH refers to hard broad forest composed of *Cinnamomum camphora*, *Quercus L.* and other hard broad species; BLS refers to soft broad forest composed of *Populus* L., *Paulownia fortunei* and other soft broad species; and three types of mixed forests: mixed coniferous forest (MCF), mixed broadleaf forests (MBF) and mixed conifer-broadleaf forests (MCBF); (**c**) shows the area share of different forest patches; (**d**) shows average age composition of forest patches; (**e**) shows the origin of the forest patches (natural forest/planted forest).

**Figure 2 ijerph-19-09184-f002:**
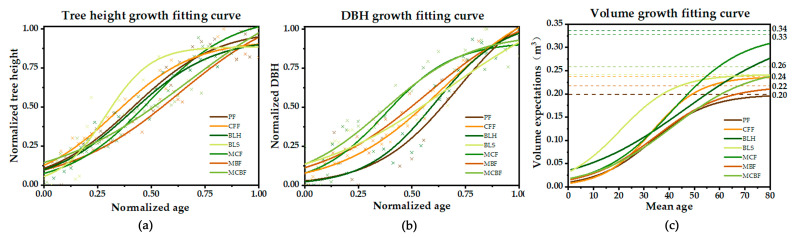
Fitting the growth model of accumulation volume of the forest types in Lushan City. (**a**) shows the tree height-standardized age fitting relationship; (**b**) shows the DBH-standardized age fitting relationship; (**c**) shows the accumulation volume expectation-mean age fitting relationship after random simulation. (**a**–**c**) compare the relative tree height, relative DBH, and accumulation volume with age curves of different forest types, respectively, which illustrates that the Logistic function has a good fitting effect and also describes the differences between the curves of different forest types, fully reflecting the growth characteristics of forest types in Lushan City. (The points in the graph were forest patches sampling in FMID data).

**Figure 3 ijerph-19-09184-f003:**
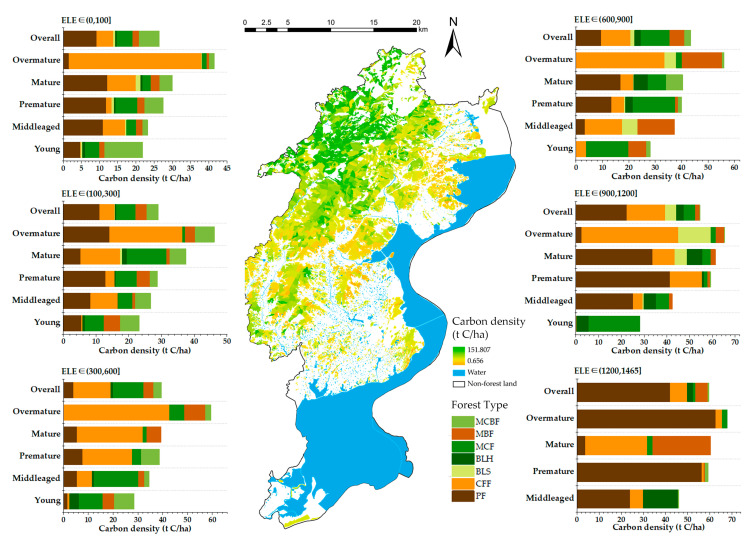
Characteristics of the current carbon sequestration capacity of forests in Lushan City. The carbon sequestration capacity of forests in Lushan City was calculated from four aspects: spatial distribution, elevation, age group, and forest types. The geographic distribution map in the middle shows the spatial distribution of carbon density of forests in 2019; the carbon density stacking figures on both sides show the carbon density share of each age group of forest types in Lushan City at (0, 100], (100, 300], (300, 600], (600, 900], (900, 1200], and (1200, 1465] altitude gradients.

**Figure 4 ijerph-19-09184-f004:**
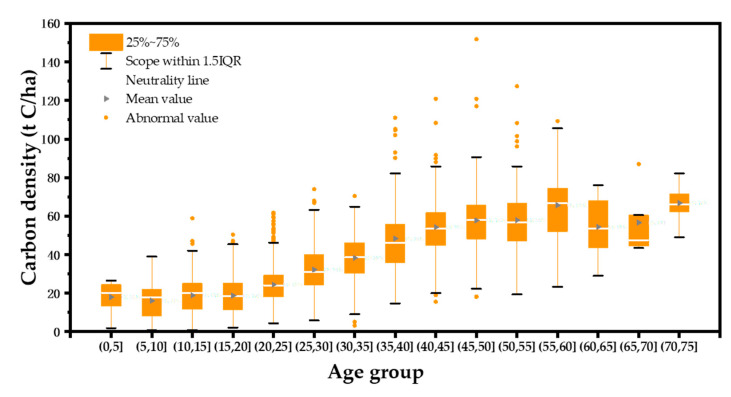
Box-and-whisker plot for forests carbon density by age in Lushan City. The figure shows the median, 25th and 75th percentile, mean (triangles), range, and extreme values outside the range (the proportion of the interquartile range past the low and high quartiles is 1.5, points outside this range will be identified as outliers).

**Figure 5 ijerph-19-09184-f005:**
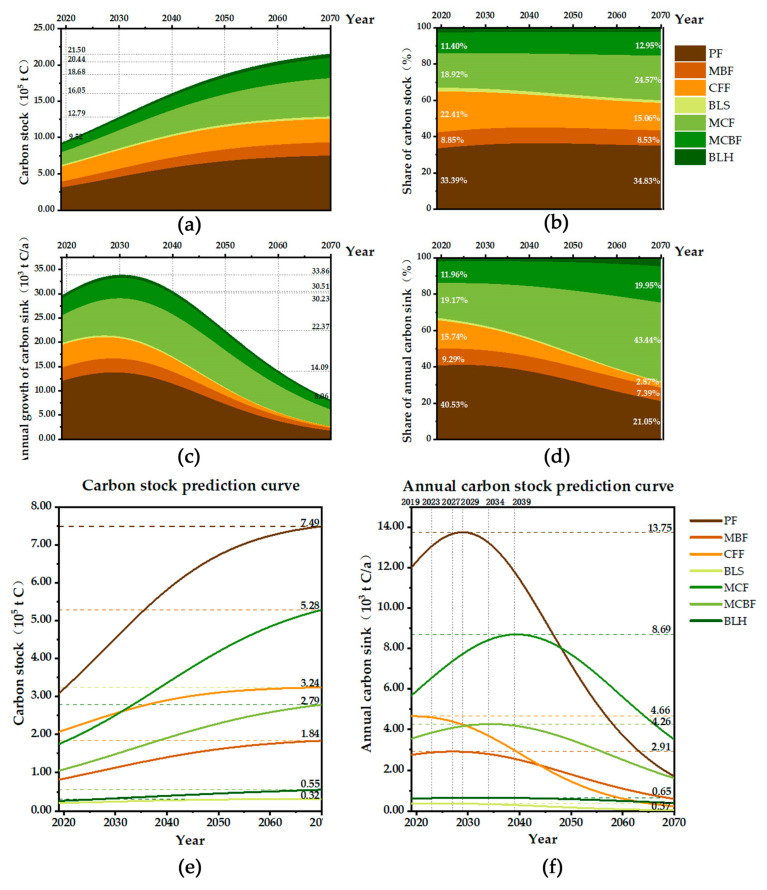
Carbon sequestration potential of different forest types in Lushan City from 2019–2070: (**a**) depicts the change of carbon stock of forest types in Lushan, (**b**) records the contribution of different forest types to the total carbon stock more visually in percentage; (**c**) depicts the change of annual carbon sink of forests types in Lushan; and the contribution of different forest types to the total annual carbon sink is visually represented in (**d**); (**e**) compares the change of carbon stock of different forest types, and (**f**) compares the annual carbon sink changes of different forest types and records the peak and arrival years.

**Figure 6 ijerph-19-09184-f006:**
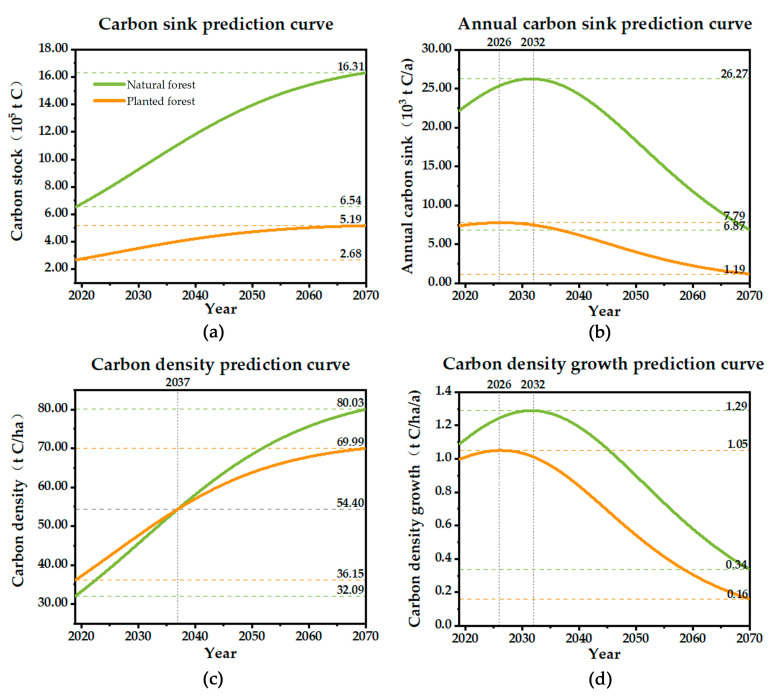
Carbon sequestration potential of natural and planted forests in Lushan City from 2019–2070: (**a**) shows the change of carbon stock, the two curves show an upward trend of decreasing growth rate, and the natural forest curve is always above the planted forest; (**b**) shows the change of annual carbon sink, the two curves show an increasing and then decreasing trend; (**c**) shows the change of carbon density, the two curves show an upward trend, and the natural forest carbon density exceeds the planted forest in 2040. (**d**) is the change of carbon density growth rate, and the change trend is similar to (**b**).

**Figure 7 ijerph-19-09184-f007:**
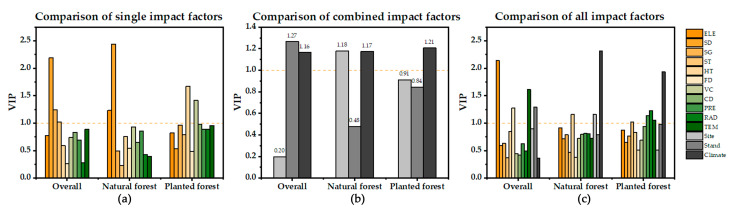
Variable Importance in Projection (VIP) for condition factors. (**a**) shows the comparison of VIP values between single factors for overall, natural forest, and planted forest; (**b**) shows the comparison of VIP values between combined features for overall, natural forest, and planted forest; (**c**) shows the comparison of VIP values between single factors and combined features. Mean elevation (ELE), Slope direction (SD), Slope gradient (SG), Soil thickness (ST), Humus thickness (HT), Forest density (FD), Vegetation cover (VC), Canopy density (CD), Precipitation (PRE), Radiation (RAD), Temperature (TEM).

**Table 1 ijerph-19-09184-t001:** BEF parameters and carbon content coefficients of forest types.

Forest Type	Main Dominant Tree Species	Model Parameters		Carbon Conversion Coefficients	
		$\beta_{1}$	$\beta_{0}$	γ	References
PF	*Pinus massoniana*	0.520	0	0.544	[40]
	Other pine species such as *Pinus elliottii*	0.517	33.238		
CFF	*Cunninghamia lanceolata*	0.399	22.540	0.555	[40]
	*Cryptomeria japonica*, *Metasequoia glyptostroboides*	0.416	41.332		
	*Cupressus funebris*	0.613	26.145		
BLH	*Quercus* L.	1.145	8.547	0.522	[40]
	*Cinnamomum camphora*	1.036	8.059		
	Other hard and broad categories	0.756	8.310		
BLS	*Populus* L., *Paulownia fortunei* and etc.	0.475	30.603	0.521	[40]
MCF	-	0.589	24.515	0.528	[39]
MBF	-	0.839	9.416	0.511	[39]
MCBF	-	0.802	12.280	0.494	[39]

PF refers to pine forest; CFF refers to Chinese fir forest; BLH refers to hard broad forest; BLS refers to soft broad forest; and three types of mixed forests: MCF refers to mixed coniferous forest, MBF refers to mixed broadleaf forests, and MCBF refers to mixed conifer-broadleaf forests.

**Table 2 ijerph-19-09184-t002:** Selection of single-factor and combined characteristics of the conditional factors.

Portfolio Feature Type	Single Factor	Portfolio Features
Site Characteristics	Mean elevation (ELE), Slope direction (SD), Slope gradient (SG), Soil thickness (ST), Humus thickness (HT)	$\sqrt[{5}]{\prod_{j=1}^{5}{\tilde{X}}_{1j}}$
Stand Characteristics	Forest density (FD), Vegetation cover (VC), Canopy density (CD)	$\sqrt[{3}]{\prod_{j=1}^{3}{\tilde{X}}_{2j}}$
Climate Characteristics	Precipitation (PRE), Radiation (RAD), Temperature (TEM)	$\sqrt[{3}]{\prod_{j=1}^{3}{\tilde{X}}_{3j}}$

$\tilde{X}$ is the normalization of single factor X in the table.

**Table 3 ijerph-19-09184-t003:** Fitting results of the volume growth models in different forest types.

Model Type		Forest Type						
		PF	CFF	BLS	BLH	MCF	MBF	MCBF
Binary standing volume model	a_0_	7.695 × 10^−5^	6.445 × 10^−5^	7.199 × 10^−5^	9.161 × 10^−5^	7.642 × 10^−5^	1.086 × 10^−4^	1.081 × 10^−4^
	a_1_	1.953	1.939	1.953	1.981	1.969	1.884	1.892
	a_2_	0.821	0.906	0.898	0.757	0.821	0.767	0.758
	R^2^	0.934	0.957	0.891	0.906	0.932	0.933	0.950
	RMSE	0.013	0.022	0.039	0.035	0.012	0.018	0.012
DBH growth model	c_0_	1.059	0.972	0.917	1.237	1.237	1.170	1.243
	c_1_	44.533	6.218	10.829	8.026	14.709	9.276	42.803
	c_2_	6.249	4.929	6.345	3.135	4.193	3.920	5.265
	R^2^	0.898	0.845	0.853	0.855	0.973	0.926	0.799
	RMSE	0.107	0.110	0.119	0.317	0.045	0.076	0.130
Tree height growth model	c_0_	0.997	0.927	0.887	0.940	1.086	1.177	1.317
	c_1_	8.295	6.262	14.930	8.592	13.439	11.052	8.106
	c_2_	5.119	5.496	9.421	5.279	5.253	3.823	3.143
	R^2^	0.941	0.915	0.881	0.937	0.969	0.968	0.910
	RMSE	0.076	0.084	0.118	0.074	0.051	0.051	0.072
Model for growth of average plant volume	c_0_	0.199	0.237	0.242	0.336	0.328	0.217	0.258
	c_1_	19.486	30.880	6.591	8.510	18.997	13.711	13.693
	c_2_	0.089	0.104	0.092	0.046	0.072	0.075	0.063
	R^2^	0.991	0.997	0.988	0.999	0.996	0.994	0.995
	RMSE	0.006	0.003	0.006	0.002	0.006	0.005	0.005

PF refers to pine forest; CFF refers to Chinese fir forest; BLH refers to hard broad forest; BLS refers to soft broad forest; and three types of mixed forests: MCF refers to mixed coniferous forest, MBF refers to mixed broadleaf forests, and MCBF refers to mixed conifer-broadleaf forests.

**Table 4 ijerph-19-09184-t004:** Status of carbon density/carbon stock in forest patches of different types.

Forest Type	Forest Volume (×10^4^ m³)	Forest Volume Density (m³/ha)	Tree Biomass (×10^4^ t)	Forest Carbon Stock (×10^4^ t)	Forest Carbon Density (t/ha)	Forest Single-Year Carbon Sink (×10^3^ t/a)	Forest Carbon Intensity Growth (t/ha/a)
PF	89.617	78.007	56.558	30.784	26.796	12.291	1.070
CFF	60.795	123.397	37.211	20.660	41.933	4.653	0.944
BLH	4.777	91.650	5.008	2.615	50.181	0.614	1.179
BLS	4.976	99.408	3.897	2.029	40.543	0.372	0.744
MCF	35.161	70.012	33.022	17.442	34.730	5.893	1.173
MBF	16.705	80.809	15.966	8.159	39.457	2.784	1.347
MCBF	22.072	75.560	21.287	10.509	35.976	3.619	1.239
Overall	234.103	85.291	172.948	92.197	33.590	30.227	1.101

PF refers to pine forest; CFF refers to Chinese fir forest; BLH refers to hard broad forest; BLS refers to soft broad forest; and three types of mixed forests: MCF refers to mixed coniferous forest, MBF refers to mixed broadleaf forests, and MCBF refers to mixed conifer-broadleaf forests.

**Table 5 ijerph-19-09184-t005:** Comparison with the estimated and predicted values of forest carbon density in Jiangxi Province from previous studies.

Study Area	Survey Time	Status Quo Carbon Density of Different Forest Types (t C/ha)							Average Carbon Density (t C/ha)	Predicted Year Carbon Density (t C/ha)				References
		PF	CFF	BLH	BLS	MCF	MBF	MCBF		2020	2030	2040	2050	
Lushan City	2019	26.8	41.93	50.18	40.54	34.73	39.45	35.98	33.59	34.69	46.61	58.49	68.04	This study
Taihe County, Jiangxi Province	2003	*Pinus massoniana* 13.76 *Pinus elliottii* 37.8	29.09	32.46	33.68	27.79			26.31	35.91	40.37	-	-	[55]
Xingguo County, Jiangxi Province	2003	*Pinus massoniana* 13.28 *Pinus elliottii* 36.89	24.65	59.96	44.23	44.94			18.25	-				[56]
The whole of Jiangxi	2001–2005	*Pinus massoniana* 14.89 Foreign pine 37.68	29.51	42.64	32.3	33			27.2	-				[52]
The whole of Jiangxi	2011	*Pinus massoniana* 9.69 *Pinus elliottii* 8.49	20.77	16.18	21.25	27.05	35.46	26.25	23.87	-				[53]
The whole of Jiangxi	2016	34.54	33.16	-	-	43.11	54.51	40.69	36	-				[24]
The whole of Jiangxi	2013	-							28.95	30.39	-	-	40.55	[54]
Entire Jiangxi/National	2010	-							20.68	41.76	45.81	48.55	52.52	[58]
National	2010	-							-	50.51	58.17	63.73	67.84	[46]
National	2000	-							-	59.8	65.1	68.9	71.7	[16]

PF refers to pine forest; CFF refers to Chinese fir forest; BLH refers to hard broad forest; BLS refers to soft broad forest; and three types of mixed forests: MCF refers to mixed coniferous forest, MBF refers to mixed broadleaf forests, and MCBF refers to mixed conifer-broadleaf forests.

## Data Availability

Not applicable.

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
