# Peer review of "Estimating Carbon Sequestration Potential of Forest and Its Influencing Factors at Fine Spatial-Scales: A Case Study of Lushan City in Southern China"

_ijerph, 2022, doi:10.3390/ijerph19159184_

Round 1

Reviewer 1 Report

The paper brings research about carbon sequestration potential (prediction) at Southern China taking in account trees specific aspects. The issue deserves attention due to environmental preservation.

In Abstract there is no mention which contribution is made on the paper.

The Introduction mentions several details making clear the theme will be treated. However, the lack of deep literature review doesn't show what the contribution of this paper related the past similar articles.

Figure 1: how the information about “Forest type” were defined?

Eq 1: Please check Y letter

Please describe better the Equation 2

Eqs 3-6: please describe how the equations were achieved.

Author Response

We would like to thank you for careful and thorough reading of this manuscript and for the thoughtful comments and constructive suggestions, which help to improve the quality of this manuscript. We have made a revision of this manuscript based on your comments and suggestions. We respond to the reviewer’ specific questions and concerns below (the reviewer’ comments are in bold and italics).

Reviewer 2 Report

    The paper by He et al titled “Estimating carbon sequestration potential and its influencing factors of forest at fine spatial-scales: a case study of Lushan City in Southern China” estimate the region carbon sequestration potential by analyzing the current characteristics of carbon stock, and reveal the influence of condition factors and their combination characteristics (site, stand, and climate) on carbon sequestration potential without forest growth disturbances, which is significant for carbon peaking and carbon neutrality. However, some problems are existed in the research and writing. 

The authors should carefully address the comments below:

1. P3, line 103-107, the reviewer strongly recommend merging into the “study area”. The introduction is more concerned with raising scientific issue and how to study them, rather than emphasizing the representation and importance of study area.

2. P5, line 163-164, the radiation data in this study are high temporal resolution, but do not have high spatial resolution. Therefore, it is recommended to revise to high temporal resolution data.

3. Is it accurate to resample the radiation data from 10 km to 30 m? It is recommended to use fishnet extraction to extract raster data, and then use kriging difference to generate 30 m radiation data.

4. Solar radiation plays an important role in plant carbon sequestration, for example, the plant carbon sequestration capacity on sunny slopes may be higher. However, sunny slopes lead to strong soil mineralization and evapotranspiration, which may limit plant carbon sequestration. Since the authors did not measure soil nutrient mineralization and evapotranspiration, it is recommended that a statement of limitation be added to the uncertainty.

5.P20, line 554-555, the conclusion is not the characteristics of the study area, but the general law. Therefore, it is recommended to delete keywords such as the study area in the conclusion, and modify it to subtropical coniferous and broadleaf mixed forest. For example, by 2070, carbon sequestration of subtropical mixed coniferous forests will reach an approximate maximum.

6. Verifying references. Some of the journal titles are abbreviated, while others are full.

Author Response

(The authors gave the same response as above.)

Reviewer 3 Report

The article deals with the carbon sequestration potential of forests in a selected region of China, with some of the issues (the methodological ones) potentially having wider applicability. The article is of great practical relevance as it may help to set at least local policies for carbon sequestration and forest management. However, it requires additions and corrections. I provide detailed comments on the article below.

Title

I propose to move the words "of forest" behind the word "potential"

Abstract

L16 - unnecessary word "And" at the beginning of a sentence

L20 - please review the comments on this date later in the review

The third result - not very correct sentence construction, making it difficult to understand

Introduction

L36 - in what sense 'due to climate change'? please explain

Materials and Methods

Subchapter 2.1 - it would be useful to add information on how long the nature reserve has been in operation and what previously happened to the forests in the area. If "the current vegetation is secondary vegetation in natural recovery", how advanced is this process (how much do the current forests resemble those that should grow here naturally)? Additionally, no sources of information have been provided.

L147 - do 'economic forests' not include trees, that they have been excluded from the analysis?

L149 - instead of 'class survey samples', can't you just write patches? this will tie in better with the previous sentence. Were all these 76 factors analysed by the authors?

L157 - unexplained term "natural forests" - are these all patches taken for analysis? However, it was mentioned earlier that "the current vegetation is secondary vegetation", so I'm not sure if this can be so explicitly described as "natural forests". Disturbances that were before "natural recovery" may have negatively affected habitat fertility, for example, and "natural recovery" will not then lead to forests as they were before the disturbance... At the same time, I see further in the text the phrase "plantation forests" - it certainly no longer fits with the earlier information that these forests are "secondary vegetation in natural recovery". Please correct the description so that there is no doubt about the quality and history of the forests included in the study.

L157-158 - "one age class limit". - it is worth adding how many years specifically

L159 - "and this study travelled to the field" - probably more like the authors of this study travelled to the field....

L177 - instead of Y it should be V

L197-198 - no explanation of the letter 'e' in the formula

L205 - instead of 'this study' rather 'authors'

L219 - why is the unit hm2 used instead of ha? throughout the text (and in Figures) this should be corrected

Eq.6 - isn't this formula more suited to determining the difference over some general period (from 0 to t), whereby for it to be 1 year, the current formula should probably be divided by the number of years of that period, i.e. t? Conversely, if Eq.6 were to reflect only 1 year, it would be less confusing to use the indices 0 and 1 or t-1 and t

Table 1 - I think there are some offsets in the column headings. What is meant by 'advantegeous' tree species? Are these species not there but advantegeous would be?

L240-241 - why isn't a complete list of the factors studied given? if it's too long to give in the text, you could always make it an appendix

Table 2 - shouldn't SG have some broader explanation beyond the word "slope" (there is a Slope gradient in the results - the description under Figure 7 should also be completed)? Why "planting density" when it was previously stated that the forests studied are "secondary vegetation in natural recovery"? "depression" - in what sense does this refer to "stand characteristics"? because depression to me connotes a depression in the ground, which would fit more with site characteristics

The description of the forests still lacks information that is relevant for understanding/interpreting the results, e.g. what is the overall age structure of the analysed forests, what is the proportion of natural versus planted forests.

Results

L267 - "for each dominant tree species" - probably more like "for each type of forest"?

Figure 2 - the individual graphs are too small and thus unreadable. Perhaps this could be broken up into 3 separate figures rotated by 90⁰? It also seems to me that the descriptions of the vertical axes in part b should be different (Normalized tree DBH). Caption to Figure 2: L283 - forest types rather than dominant tree species (it wasn't previously stated to be otherwise).

L291 - is this the result of the authors' calculations or information from FMID? This is worth clarifying (I see further that this result is in Table 4, so can be referred to)

L293-295 - "different age groups". - in the description of the methodology, it should be added what "age groups" have been separated and what range of age/years goes into each group. "Of dominant tree species" - the description also refers to forest types, not individual tree species, this should be added in this sentence and generally elsewhere in the article (e.g. in L299: "The main contributing tree species for carbon density was CFF". - after all, CFF is not a species, but a forest type composed of at least 4 species - vide Table 1)

L296 - "and the carbon density was 26.41-28.97 t/hm²" - you might want to add "here" as it does not follow from this part of the sentence

L299-300 - "The main contributing tree species for carbon density was CFF in the 300-600 m and 600-900 m gradient intervals" - it seems that this contribution for these altitudes is very similar for MCF as well (especially in the 600-900 m gradient) - perhaps worth adding?

L306 - "dominant species forest patch" -> "forest type"; "the four forest patches" - these patches were rather a lot more than "four"

L308 (and onwards) - "will be" or "was"? (the description refers to the years 2019-2020, which have already passed). Also missing in this paragraph is a reference to Table 4

L310 (and elsewhere in the article) - "1.10 t/hm2*a-1" - it seems to me that the notation "1.10 t/ha/y" or possibly "1.10 t/ha/a" would be clearer

Figure 3 - graph for ELE 900-1200 - the x-axis is signed 'Mg C/ha', all others give 't' as the basic unit. Caption for Figure 3 - is the word "tree" before "forests" needed? "the carbon density stacking maps on both sides" - these are not maps, but graphs next to the map. There are also unnecessary repetitions of text and incorrect height ranges in the following text: "the carbon density share of each age group of the main dominant tree species types in Lushan City at (0,100], (100,300], (300,600], (600,900], (900 The carbon density share of each age group at (0,100], (100,300], (300,600], (600,900], (900,900], (1200], (1200,1465], and (1200,1465] altitude gradients".

Table 4 - in the title, instead of "dominant tree species", I propose to write "forest types"

L325 (and further) - tree forests -> forest types?

Subchapter 3.3 - there are no forest patches older than 75 years in the area? this should be added in the text, because maybe if they were older, the upper limit of carbon density would be shifted?

L327-329 and Figure 4 - "this study concludes that the carbon stock of Lushan City will converge to the maximum in 2070". - on what basis do the authors make such a claim? did they analyse the changes in the area of the age classes of the current forest stands up to 2070 and conclude that at this age (55-75 years) the relatively largest area of the current forests will be found? well, after all, at the moment this area varies in terms of age classes, so to conclude simply from Figure 4 that in 50 years there will be the highest carbon density is not correct. Unless Figure 4 is incorrectly described and the X axis is not the average ages of the stands but the projected periods (dates): 0-5 -> 2020-2025, 5-10 -> 2025-2030 etc. However, the way I understand it is that Figure 4 should only be a starting point for further simulations, when the maximum of carbon stock/density may occur in the future, based on the known current forest stand layout (their area in each age range). Are these forests currently subject to any use or conservation measures that could affect the amount of carbon sequestration?

L330 - there should be a reference to Figure 5 here

L355 - in subchapter 2.2 it refers to 'plantation forests', here it refers to 'planted forests'. Obviously every plantation forest is also a planted forest, but not every planted forest is also a plantation forest - the latter implies intensive timber production, so an management model quite distant from natural forest and the vast majority of planted forests....

L358 - intensity or density?

L360 - carbon density (not alone "density"), additionally in Figure 6 it is 80.03 and in the text 80.02

L362 - more sustainable or rather more effective?

Discussion

L428-435 - I do not really understand the presence of this fragment. After all, the authors did not describe in the methodology the fact of comparing the NPP data and the estimated carbon stock status (the Pearson correlation test). There is also no reference for the MODIS product or information on how it works.

L440 - forest type rather than species

L440-443 - I am not convinced that these results from the first three rows of Table 5 for individual forest types can be called similar (especially rows 1 and 2). Moreover, only in some cases the results of the authors of the article are higher than the results of row 2 and 3....

L446 - it would be appropriate to add that reference [24] refers to the whole of Jiangxi Province, not just Lushan City.

L453 – why „trend”? maybe „data”?

L456 – rather „trees” than „people”

Table 5 - title - seems like it could be simplified. Column headings - units should be corrected to "t C/ha"

L471-473 - this sentence seems to be unfinished

L476-478 - it would be appropriate to add that in China

Subchapter 4.2 - it would be worth expanding it further with more specific results from the authors as to under which conditions of site/stand/climate, carbon sequestration is greatest - this could be a hint for afforestation programmes (where and how it is most worthwhile to afforest so that forests will sequester carbon to the greatest extent). I also note the term plantation forest used (is it correct?).

L507-508 - repetition in part of sentence: "all of which increased the forest's carbon sequestration potential [61], and thus improved forest carbon sequestration potential [5]".

Subchapter 4.3 - it is worth adding wording to state that the calculations made by the authors refer to the current forest area and specifically identify its potential development, without taking into account planned afforestation

Conclusions

L554 - to be revised in line with my earlier comments

L561 - add 'gradient' to the single word 'slope'. And vegetation cover?

L565 - probably unnecessary word "be"

References

Mostly articles by Chinese authors are quoted, wouldn't there also be some interesting items from other regions of the world?

Author Response

(The authors gave the same response as above.)

Round 2

Reviewer 1 Report

Dear,

Thanks for you latest revision that showed good progress. However, I am afraid there are still a few minor things to strighten out. Like as Figure 2 and 3 due the size of page.

Author Response

Thank you for careful reading and thoughtful comments, which help to improve the quality of this manuscript. We have updated the phrasing in the entire text and made a revision of this manuscript based on your comments. In addition, we have reinserted figures that fit the page size to make them clearer. Please review it in the revised manuscript.

Reviewer 3 Report

Thank you very much for your extremely careful, precise and professional response to my review and the changes made to the article. Below are my final comments on the revised version, line numbers refer to the version of the article with all corrections visible.

Abstract

The second to last sentence is redundant, repeating previous information. Last sentence needs linguistic correction

Materials and Methods

In Subchapter 2.1 it would be worth adding whether there are any activities (conservation work, removal of parts of trees) in these forests that may affect carbon sequestration.

L156 - isn't the word 'strong' a mistake? as age classes were supposed to be given here - maybe you mean middle-aged trees?

L159-161 - this sentence should be removed (it goes on, in the revised version)

L184-185 - "Excluding economic forest, shrub forest, bamboo forest and other forest patch types" - it is worth adding information here in brackets as to why they were excluded (not enough data or no trees etc?).

Results

L366 - probably unnecessary word 'patch'

L391-392 - "will be between 55 and 75 years" - it may be worth adding here that this is the average age of the forest stands, as previously the numbers are used in a different context

Discussion

Table 5 - I was not suggesting to remove the results of these two local surveys, I was just pointing out that the textual commentary on them is not very precise and compatible with the results in this table

Conclusions

L651 - I suggest instead of "forest" to give "forests", it will fit better in the next sentence

App. 1

Origin 2018 is cited and in the article it is Origin 2019b

Author Response

Thank you for careful reading and thoughtful comments, which help to improve the quality of this manuscript. We have made a revision of this manuscript based on your comments. We respond to the reviewer’ specific questions and concerns below (the reviewer’ comments are in bold and italics).
